# Carbohydrate Ingestion before Exercise for Individuals with McArdle Disease: Survey Evidence of Implementation and Perception in Real-World Settings

**DOI:** 10.3390/nu16101423

**Published:** 2024-05-09

**Authors:** Sam L. Torrens, Evelyn B. Parr, Craig McNulty, Lynda Ross, Helen MacLaughlin, Robert A. Robergs

**Affiliations:** 1School of Exercise and Nutrition Sciences, Faculty of Health, Queensland University of Technology, Victoria Park Road, Kelvin Grove, QLD 4058, Australia; c.mcnulty@qut.edu.au (C.M.); l20.ross@qut.edu.au (L.R.); h.maclaughlin@qut.edu.au (H.M.); rob.robergs@qut.edu.au (R.A.R.); 2Mary Mackillop Institute for Health Research, Australian Catholic University, Level 5, 215 Spring Street, Melbourne, VIC 3000, Australia; evelyn.parr@acu.edu.au

**Keywords:** McArdle disease, patient experience, carbohydrate, sucrose, exercise, clinical practice guidelines

## Abstract

In individuals with McArdle disease (IWMD), the ingestion of carbohydrates before exercise has previously been shown in laboratory studies to significantly decrease the exercising symptoms of the condition and increase exercise tolerance during the early stages of exercise. As a result, carbohydrate ingestion pre-exercise is currently included in management guidelines, and often advised by medical professionals treating the condition. The aim of the current study was to determine whether positive lab-based results for the ingestion of carbohydrate before exercise in laboratory studies are being effectively translated into practice and produce perceptions of the same positive outcomes in real-world settings (RWS). An online survey method was used to collect responses from 108 IWMD. Data collected on the amount and type of carbohydrate consumed prior to exercise found that most surveyed participants (69.6%) who supplied qualitative data (*n* = 45) consumed less than the 37 g currently recommended in management guidelines. Survey data also revealed a large variation in the type and amount of carbohydrate ingested when IWMDs are applying carbohydrate ingestion before exercise in RWS. Consistent with these findings, only 17.5% of participants stated that they found carbohydrate ingestion before exercise relieved or minimised their MD symptoms. Results suggest that positive lab-based findings (increased exercise tolerance) of carbohydrate ingestion before exercise are not being effectively translated to RWS for many IWMD. There is a need for improved patient education of IWMD on the application of carbohydrate ingestion before exercise in RWS.

## 1. Introduction

McArdle disease (MD) is a rare (1:100,000) genetic myopathy whereby individuals are unable to produce the enzyme glycogen phosphorylase in skeletal muscle tissue. Muscle glycogen phosphorylase catalyses the first step of glycogenolysis through phosphorylation from muscle glycogen to glucose-1-phosphate. As a result, stored muscle glycogen cannot be accessed to fuel glycolysis and subsequent oxidative phosphorylation within the mitochondria. The inaccessibility to muscle glycogen causes a reduction in the capacity for energy release from carbohydrate oxidation to support muscle contraction and other chemical-driven systems of cellular work. Therefore, individuals with MD (IWMD) heavily rely on alternative substrates such as fatty acids, liver glycogen, and blood glucose to fuel the cellular energy demands of skeletal muscle during activities of daily living, as well as exercise, for health promotion and disease prevention [1,2]. As muscle glycogen plays an integral role in mediation between anaerobic and aerobic cellular ATP resynthesis, deleterious symptoms (increased HR, increased RPE, increased muscle stiffness, cramping, contractures, decreased exercise tolerance) are most notable in IWMD during rapid increases in cellular ATP demand, such as during the early stages of exercise and abrupt increases in exercise intensity.

For non-MD individuals, ATP resynthesis during the first few minutes of exercise is predominantly met by the breakdown of creatine phosphate and muscle glycolysis fuelled by intramuscular glycogenolysis [3,4]. As IWMD do not have muscle phosphorylase (hence no access to muscle glycogen stores) and creatine phosphate is limited in its supply, times of rapidly increased metabolic demand result in a cellular energy crisis due to the lack of available substrate for rapid ATP resynthesis. As a result, IWMD must wait for alternative fuel sources to arrive at the working muscle for subsequent breakdown and ATP resynthesis to meet the increased metabolic demand. The alternative fuels (free fatty acids and liver-derived blood glucose) must first be mobilised from their site of origin and then transported to the active muscle, which results in the delay of their utilisation for ATP resynthesis. Once these alternative fuels reach the active muscle, IWMD can replace the substrate demand that would normally (in non-MD individuals) be met by muscle glycogen, though the capacity and rate response of their muscle metabolism remains relatively constrained compared to non-MD individuals. Hence, IWMD see a dramatic decrease in symptoms after ~6–8 min of activity. This phenomenon, termed the MD ‘second wind’ [5] is a distinct feature of the condition. However, the term ‘second wind’ is somewhat of a misnomer. This is because the response is unrelated to ventilation and/or external respiration and results from a systemic metabolic adjustment whereby the additional substrate becomes available to the working muscle.

Early research studies [6,7] discovered that the MD metabolic adjustment (second wind) could be prematurely induced during the early stages of exercise when participants were infused via cannula with glucose for 30 min before the initiation of exercise. By infusing glucose 30 min before exercise, researchers were able to prime the individual with free plasma glucose that could be immediately utilized by the exercising muscle at the onset of exercise (hyperglycaemia). The infused carbohydrate could then enter the working muscle via its conversion into glucose-6-phosphate (hexokinase reaction, thereby bypassing glycogenolysis), increasing the supply of available substrate for ATP production.

While direct glucose infusion was an effective measure to induce hyperglycaemia and elicit a MD metabolic adjustment during the early stages of exercise [6,8], its application to real-world settings (RWS) was unrealistic due to the need for venous cannulation and a constant infusion of carbohydrate. In order to provide a more practical alternative, subsequent research focused on the oral ingestion of simple carbohydrates (sucrose) to elicit hyperglycaemia prior to exercise and induce an MD metabolic adjustment [9,10], producing results similar to that of the previous infusion studies (i.e., improved exercise tolerance).

The ingestion of 37 g of sucrose 5–10 min before exercise has been advised in the most recent MD clinical management guidelines [11] and often advised as an initial management method for the condition by medical professionals. However, despite research [9,10] demonstrating consistent and significant benefits when employing carbohydrate ingestion prior to exercise in research settings, the translation of these positive results into RWS has never been assessed. Consequently, the purpose of this research was to determine whether the management technique of carbohydrate ingestion before exercise, advised in the most recent clinical management guidelines, is of benefit to those IWMD employing it in RWS to relieve and minimise their MD symptoms during exercise.

## 2. Materials and Methods

With custodian consent, an online survey (conducted via Qualtrics, Seattle, WA, USA) was advertised on the International Association for Muscle Glycogen Storage Disease’s (IamGSD) community Facebook page, an MD patient advocacy group. The survey was available for completion for a 7-week period (from 17 March 2022) and interested participants were invited to take part in capturing individual responses. The survey was approved by the Queensland University of Technology Ethics Committee (approval number 4583) and participants were required to be 18 years of age or older and have a genetic diagnosis of McArdle disease.

The survey comprised twenty-seven questions including a broad range of topics including participant characteristics, exercise habits, the use of carbohydrates before exercise as a MD management technique, daily activity, medical history, disease impact, disease education, and access to disease-related educational resources (see Appendix A). Participants were not required to complete all questions available on the survey for their responses to be recorded. Seventeen questions were multiple choice where only a single option could be selected; two questions were multiple choice where one or more options could be selected for the same question; seven were short response questions; and one was a multiple choice question, where more than one option could be selected with the option of a short response. Open text boxes were available to capture short responses.

### Data Analysis

For the purpose of this manuscript in which we specifically investigated the implementation and perceptions of carbohydrate ingestion before exercise of IWMD in RWS, questions referring to participant characteristics, exercise habits, and the use of carbohydrates before exercise as a MD management technique were utilized for analysis (nine in total). Specific questions utilized are labelled in Appendix A. Where participants were asked to report the type and amount of carbohydrate ingested before attempting exercise, participants could enter multiple responses and sufficient carbohydrate ingestion was determined as a total amount of carbohydrate intake equal to or exceeding the 37 g of carbohydrate currently advised in the most recent clinical practice guidelines. Results are presented as totals.

## 3. Results

A total of 108 participants provided survey responses. Because participants were not required to complete all questions available on the survey for their responses to be recorded, the total number of responses for each question varied as not all participants elected to answer all questions provided.

A summary of participant details is presented in Table 1.

Questions 10 and 11 asked participants about their continuous physical activity participation. The responses of 94 participants are detailed in Table 2. Among 104 survey responses to question 10 asking the number of days they participated in continuous physical activity, 10 participants selected that they do not undertake any continuous physical activity and therefore recorded no response for question 11.

In response to survey question 15. “Do you find the current McArdle disease management guidelines of consuming sugary drinks or foods before you are physically active relieves/minimises your McArdle symptoms”, of 103 responses, only 17.5% of participants confirmed that the management technique of pre-exercise carbohydrate ingestion relieved or minimised their MD symptoms (Figure 1). The majority (39.8%) of participants confirmed that pre-exercise carbohydrate ingestion did not relieve or minimize their MD symptoms. When those who answered ‘Sometimes’ were combined with those who answered ‘Yes’, this represented less than half the participants surveyed (46.6%), experiencing at least one positive event where carbohydrate ingestion before exercise was of benefit (Figure 1). The remaining 13.6% of participants had never tried carbohydrate ingestion before exercise.

The reported sources of carbohydrate utilised (Figure 2) and the amounts ingested (Figure 3) when employing the technique of carbohydrate ingestion before exercise displayed large variation. Fruit or fruit juice and sports drinks were the most reported carbohydrate sources ingested prior to exercise (Figure 2). Where participants were asked to report the type and amount of carbohydrate ingested before attempting exercise, 45 participants reported a total of 79 exercise attempts (participants could record more than one exercise attempt), where most attempts (69.6%) occurred with the participant consuming less than the 37 g of carbohydrate currently advised in the management guidelines (Figure 3).

## 4. Discussion

The current investigation sought to provide further understanding and detail into the perceived effectiveness of the currently advised MD management technique of carbohydrate ingestion before exercise in RWS. Participants provided survey responses to questions of their exercise habits and their use of carbohydrate before exercise to decrease their MD symptoms during exercise.

The major findings of this study were that, for many surveyed IWMD, the details of pre-exercise carbohydrate ingestion were: (1) not well understood and/or not applied in RWS as per the current guidelines; and (2) the positive lab-based results (increased exercise tolerance/reduction in MD symptoms) occurring after carbohydrate ingestion were not being experienced. This lack of translation should be of concern to researchers, policy makers, and medical professionals working with IWMD as the advice currently being given is not benefiting a large portion of those it is intended to support. Regarding item 1 above, there was a large variation in the type and amount of carbohydrate ingested when surveyed IWMD applied the technique of carbohydrate ingestion before exercise in RWS (Figure 2 and Figure 3). The large variation in the type and amount of carbohydrate ingested when applying the technique may explain the poor outcomes in RWS. Consequently, there is a need for improved patient education about the practice of pre-exercise carbohydrate ingestion for IWMD as well as further research investigation into the most effective strategies for its implementation.

There are likely multiple factors contributing to the poor translation of positive lab-based research results to perceptions of the effectiveness of carbohydrate ingestion before exercise in RWS; however, it is acknowledged that carbohydrate ingestion before exercise in IWMD has received little investigation to date. It is often stated in MD management advice that the ingestion of carbohydrate before exercise to improve exercise tolerance is ‘well established’. However, to our knowledge, there are only six published studies to date investigating the effects of orally ingested carbohydrates on exercise tolerance in MD populations [9,10,12,13,14,15], and only two of the six studies are utilised in the current clinical management guidelines. MD is a rare condition and the difficulties in completing research on such rare populations are obvious; however, this does not decrease the need for multiple studies across multiple cohorts to confirm findings before they are considered to be well established.

While there have been nine studies investigating carbohydrate infusion and exercise tolerance in IWMD [2,6,8,12,16,17,18,19,20], to group research of infusion and ingestion as one and the same would be a profound error, as both methods elicit vastly different metabolic, central nervous system, and hormonal responses [21,22]. Thus, the topic of carbohydrate ingestion before exercise in IWMD remains sparsely researched.

The large variation in how IWMD may utilise carbohydrate ingestion before exercise is illustrated by large disparities across both the amount and type of carbohydrate ingested (Figure 2 and Figure 3). While it is possible that some IWMD have found their own unique ways to ingest carbohydrate that specifically works for them based on their own body size, exercise regime, and basal blood glucose levels (BGLs), the large discrepancies suggest a lack of clear instruction/understanding for the vast majority surveyed on how to optimally ingest carbohydrate before exercise to improve exercise tolerance. This may explain the varying reports of success for carbohydrate ingestion before exercise improving exercise tolerance not only between participants, but also within individuals’ attempts at employing it (Figure 1).

To optimally utilise carbohydrate ingestion before exercise to improve exercise tolerance, an understanding of the basic mechanistic reasoning for its application should be expressed. While metabolic signalling and benefits to athletic performance (in non-MD populations) have been shown to occur at the mouth, prior to the arrival of carbohydrates in the stomach or blood stream [23,24], the mechanistic principle of the technique in regard to IWMD, is to rapidly elicit an increase in BGLs (hyperglycaemia) before exercise. The increase in BGLs allows for the exogenous carbohydrate to partially replace the metabolism of endogenous carbohydrate (muscle and liver glycogen) during the early stages of exercise, partially circumventing the blockage in glycogenolysis. This needs to be clearly understood by those employing (and advising) the technique. Therefore, to achieve this rapid rise in BGLs, the amount of carbohydrate, the type of carbohydrate, and the duration over which the carbohydrate is ingested is important. Survey results suggest that these combinations of factors are not well understood by those employing the technique, as the results from 45 participants who provided data on the type and amount of carbohydrate they consumed before exercise showed that only 30.4% attempted exercise following a carbohydrate consumption equal to or greater than 37 g of carbohydrate, as currently advised in clinical management guidelines (Figure 3). To date, this is the minimum amount of carbohydrate that has been shown in research to decrease the exercise-induced symptoms of MD. A further concern is the small proportion of individuals that reported ingesting ‘sugar-free’ beverages when trying to improve exercise tolerance, when such beverages do not contain carbohydrates that would facilitate such an outcome.

While no studies have been carried out to determine an accurate dose response (i.e., amount of carbohydrate relative to body mass), it is highly likely that the amount of carbohydrate required for successful improvement in exercise tolerance will vary between individuals. As a secondary investigation of the original oral carbohydrate ingestion study [10] where participants ingested 75 g 30–40 min before exercise, researchers had the patient with the lowest body weight (weight not specified in the research paper) undertake an additional trial, ingesting only half the original amount of carbohydrate (37.5 g of sucrose). Results found that the lower dose of carbohydrate only partially induced an MD metabolic adjustment (i.e., second wind) in comparison to fully inducing the response following the ingestion of 75 g. This indicates that there is a dose–response relationship and suggests that ingestion levels below the currently advised 37 g of carbohydrate may result in even further reductions to the induction of a MD metabolic adjustment (i.e., second wind), or completely removed any benefits to employing the technique.

Considering the research underpinning the current MD management guidelines involving carbohydrate ingestion prior to exercise suggests consuming 37 g of sucrose, it is interesting that, of the 72 participant responses, only 7 participants had actually tried the technique with 37 g of sucrose (Figure 2). If IWMD in RWS are not undertaking the same measures that showed positive results in lab studies, it is to be expected that results are not likely to be replicated. While there may be more practical substitutes to apply the technique in RWS, there is currently no research to support this. As such, currently, it would be advised that 37 g of sucrose diluted in water (method of positive lab results) is the first application applied before alternative carbohydrate sources are trialled in RWS.

The most utilized source of carbohydrate ingested when undertaking the technique was fruit or fruit juice, closely followed by sports drinks (Figure 2). Among the 72 participant responses, 35 had undertaken the technique with fruit or fruit juice, representing 48.6% of participants. This high percentage should be of interest as the ingestion of solid foods (as opposed to liquids or gels) such as whole fibrous or starchy fruit (apples, oranges, grapefruit) would delay and slow the rise in BGLs due to the additional time required for digestion and therefore would not be considered optimal options to produce a rapid spike in BGLs. Liquid or gel forms of carbohydrate would be advisable for speed of digestion, but considering the high percentage of IWMD surveyed that consumed fruit and fruit juice as the source of carbohydrate, it is important to understand that not all liquid (or solid) forms of carbohydrate are digested, absorbed, and metabolized in the same way.

Carbohydrate content within a fruit juice like apple juice primarily comprises the monosaccharide fructose (~70%), with glucose making up the remaining ~30% [25,26]. Sucrose (which is the carbohydrate advised in management guidelines) is a disaccharide and consists of one molecule of glucose (50%) and one molecule of fructose (50%) bound together. The difference in ratios of monosaccharides is important to understand as fructose and glucose have different metabolic fates once ingested. Glucose is a unique monosaccharide, because it can be directly absorbed into the blood stream from the small intestine, transported into the working muscle (insulin or contraction mediated uptake), and immediately utilized in the glycolysis pathway to provide ATP and other metabolites that can provide added ATP from complete oxidation in mitochondria [27,28]. As skeletal muscle does not possess the enzyme fructokinase which allows fructose to enter the glycolysis pathway, fructose must first be absorbed into the blood stream, transported via the portal vein to the liver (which contains the enzyme fructokinase), and then be converted to either glucose (~50%), lactate (~25%), or liver glycogen [29,30,31]. The liver-derived glucose can then be transported around the body for cellular uptake, oxidation, and ATP production. If apple juice were ingested as a substitution for sucrose diluted in water, the individual would need to drink nearly twice as much apple juice to consume and absorb the same amount of free glucose within the same time-period. This is just one example of why it should be strongly advised that IWMD undertake the technique (at least the first few attempts) with known types and quantities of monosaccharides within the beverage.

The large variation in application for the ingestion of carbohydrate before exercise is perhaps to be expected considering the guidelines around the techniques’ utilisation were not developed and published until as recently as 2021 [11]. This is nearly two decades after the original research on which the technique is founded was completed [10]. Without specific and detailed guidelines as to the appropriate steps to be taken when ingesting carbohydrate before exercise, it is likely that the large variation in its application plays a role in the lack of translation from positive lab results to RWS. The development of published guidelines for IWMD (and medical professionals) to follow and understand when implementing the technique is an important step in the management of the condition. However, current guidelines are limited in their description and instruction for optimal application, and this may be a contributing factor to poor perceived outcomes in RWS.

There is a clear need for more detailed resources explaining the principles underlying the technique of carbohydrate ingestion before exercise for IWMD and practitioners, as well as more descriptive instructions on how to optimally employ the technique. Based on the comments above, the following recommendations are founded on current scientific research and principles of exercise nutrition and physiology. In being research evidence-based, each recommendation is based on the best available evidence for the optimal benefit to IWMD in RWS.

(1)Type of carbohydrate consumed—The carbohydrate consumed when first attempting the technique should be sucrose. Sucrose is regular table sugar and can easily be diluted in water and consumed.

Current MD clinical management guidelines suggest that 37 g of sucrose approximately corresponds to one can of soda (330 mL). However, soda can have vast differences in monosaccharide ratios as well as total carbohydrate content [25]. As such, sucrose diluted in water is advised when initially utilising this technique.

The ingestion of pure glucose is also a viable and potentially more effective option than sucrose. While not directly attempting to induce a MD metabolic adjustment (second wind), Coakley and colleagues provided direct evidence of an increase in exercise tolerance following the ingestion of 75 g of glucose 20–30 min before exercise in IWMD [12,13]. Based on the research by Coakley, maltodextrin (glucose molecules connected together) should also result in substantial free glucose availability in the blood stream. However, research on maltodextrin consumption prior to exercise has yet to be completed.

Previous research in MD populations has shown increases in exercise tolerance following the infusion of sodium lactate during exercise in IWMD [6]. As discussed prior, it is also important to understand that the increased hepatic production of lactate via the ingestion, liver uptake, and subsequent metabolism of fructose derived from the sucrose disaccharide may provide a valuable substrate to the working muscle during the early stages of exercise. However, this is yet to be clearly established for ingestion protocols.

With the guidance of a medical professional, alternatives should be trialled by the individual if they have no success with sucrose or seek to improve upon the technique once success has been established with sucrose.

(2)Amount of carbohydrate consumed—The minimum amount of carbohydrate consumed should be 37 g, as any amount less than this has not yet been investigated.

It is important to note that this is not the total amount of substance to be consumed, but the total amount of pure carbohydrate to be consumed. This is why sucrose is a practical method of application as sucrose (table sugar) is pure carbohydrate. As no participant data were provided on body weight for either of the successful research trials utilising carbohydrate ingestion before exercise to induce a MD metabolic adjustment (second wind) [9,10], it can only be assumed that bigger individuals may require higher doses of carbohydrate to achieve the same benefits as smaller individuals.

(3)The consumption period—The ingestion of the carbohydrate needs to be consumed in one single bolus.

The ingestion of the carbohydrate over an extended period of time in small quantities (i.e., sipping sucrose diluted in water over the course of an exercise session) will not produce the same blood glucose response as if it were consumed all in one quantity before exercise. It is likely that the longer the time taken to ingest the total amount of carbohydrate before exercise, the less effect it will have on increasing BGLs, and as a result, likely decrease the effectiveness of the technique.

(4)The waiting/priming period prior to exercise—The optimal waiting or ‘priming’ period between the ingestion of carbohydrates and the initiation of exercise is yet to be clearly established. Nevertheless, based on past research, it would be advised that a 25–30 min waiting period is used between ingestion and the initiation of exercise.

The current recommendation in management guidelines advises a 5–10 min waiting period between ingestion and the initiation of exercise. This is based on the research by Andersen [9] in which they suggest a 5 min wait period between ingestion and exercise. This recommendation requires further research enquiry as the results of the study have never been verified by a second source, and the published data of the study suggest critical variables may have been unaccounted for during the experiments. Andersen [9] failed to acknowledge in the findings of their study that the average overnight fasted BGLs of their participants before each trial were ~7.00 mmol/L. This presents a major limitation to the results of the study as it suggests that participants were either insulin-compromised or had not followed the dietary conditions specified in testing protocols (overnight fasted). The overnight fasted BGLs reported in Andersen [9] are significantly higher than would be expected under overnight fasting conditions (normal overnight fasted BGL 4.0–5.5 mmol/L). Due to this variable, the validity of the results is severely compromised as the probability of superior results (increased exercise tolerance) after ingestion 5 min before exercise is at a significantly advantage when compared to a placebo or the ingestion of carbohydrate with a longer wait/priming period before exercise.

BGLs following the ingestion of a bolus of carbohydrate typically peak 30 min post ingestion [32]. Based on research by Kowalski, Moore, Hamley, Selathurai, and Bruce [32] and in line with previous carbohydrate ingestion research on IWMD [10,12,13], a 25–30 min wait/priming period between ingestion and the initiation of exercise is recommended until the research by Andersen [9] can be validated by a second source.

### Additional Considerations

Exercise mode and exercise intensity—The current clinical management guidelines are based upon research that was completed on stationary exercise bikes at intensities that elicited a near-maximal heart rate and level of perceived exertion before the MD metabolic adjustment (second wind) occurred. As such, the mode of exercise (i.e., cycling vs. walking vs. activities of daily living) and intensity of exercise may have a significant impact on the effectiveness of carbohydrate ingestion prior to exercise for IWMD. There is a need for continued research, education and understanding on pre-exercise carbohydrate ingestion for different types of exercise or activities of daily living.

Fasted vs. fed—The ingestion of carbohydrates in both studies showing positive effects on exercise tolerance for IWMD was completed following an overnight fast. Nutritional status prior to the ingestion of carbohydrates is also likely to influence its effectiveness as changes to nutritional status will impact liver glycogen levels [33], blood metabolite levels including blood glucose, ketone, and free fatty acids [34,35], as well as the function of the central nervous system [22].

Rebound and metabolic deficiency hypoglycaemia—It is encouraged for IWMD and medical professionals assisting them to understand the potential for the development of hypoglycaemia to occur under situations of both carbohydrate ingestion before exercise (rebound hypoglycaemia), as well as no carbohydrate intake during exercise (metabolic deficiency). While it may be argued that, because IWMD exercise at very low absolute exercise intensities, absolute carbohydrate utilisation would also be very low. However, because the relative intensity of exercise for this population is high and the only source of carbohydrate available is derived from blood glucose supplemented by the liver (due to the blockage in muscle glycogenolysis), the increased utilisation of blood glucose for muscular contraction has the potential to negatively impact BGLs and euglycemia.

As with the prescription of any management method, to prevent adverse outcomes and develop the best management method for each individual, the manipulation of BGLs should be performed under guidance and consultation with an appropriately qualified medical professional that is aware of the individual’s full medical history. The above recommendations should be thoroughly considered and manipulated for each individual and not prescribed on a one-size-fits-all basis.

## 5. Conclusions

Survey results suggest that the positive outcomes (increase in exercise tolerance/decrease in exercising MD symptoms) achieved in lab-based settings regarding carbohydrate ingestion before exercise are not being effectively translated to RWS for the vast majority of surveyed IWMD. Further research is required to determine the reasoning for this lack of transfer with potential factors including a poor understanding of the underlying principles of the management technique for optimal application by those employing and advising it, limited detail and clarity of guidelines around the utilisation of the technique, and the accuracy of prior research supporting the utilisation of the technique.

A current appraisal of this prior scientific research, in combination with the results of this survey research, revealed the need to provide clear recommendations to IWMD on how to best support the energy demands of their muscles during exercise and activities of daily living. Such recommendations should be updated as more research evidence becomes available.

## Figures and Tables

**Figure 1 nutrients-16-01423-f001:**
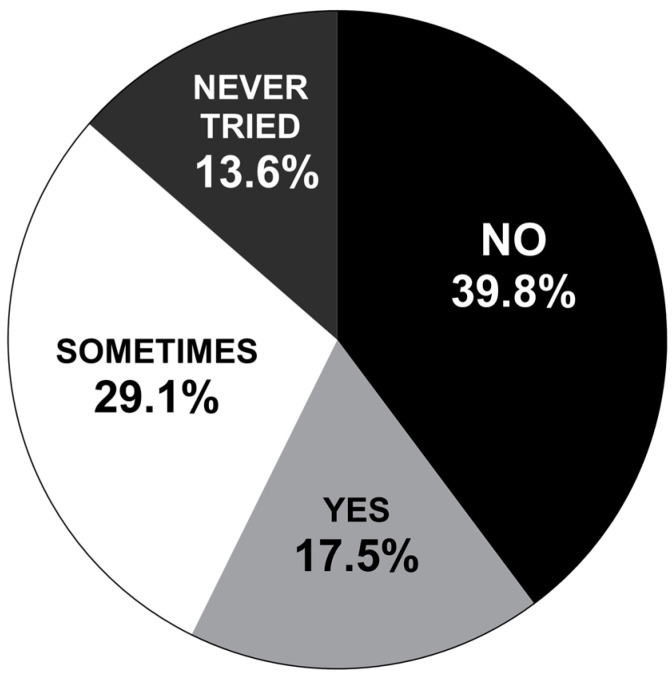
Results from 103 individuals with McArdle disease (IWMDs) when asked “Do you find the current McArdle disease management guidelines of consuming sugary drinks or foods before you are physically active relieves/minimises your McArdle symptoms?”

**Figure 2 nutrients-16-01423-f002:**
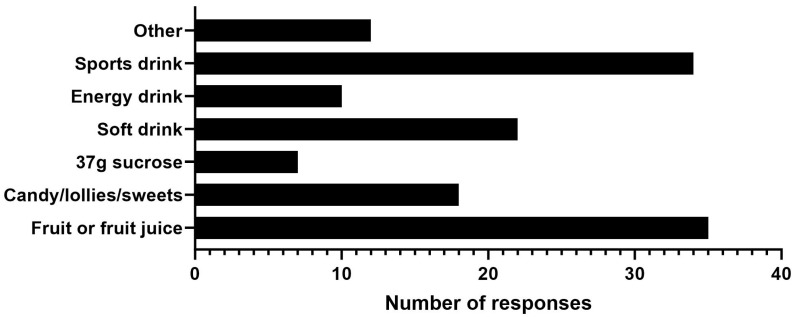
Results from 72 individuals with McArdle disease (IWMD) when asked “With regard to the current McArdle disease management guidelines of consuming sugary drinks or foods before you are physically active, what sugary drinks or foods have you tried before exercise?” Individual participants could enter multiple responses to this question (138 total responses were recorded).

**Figure 3 nutrients-16-01423-f003:**
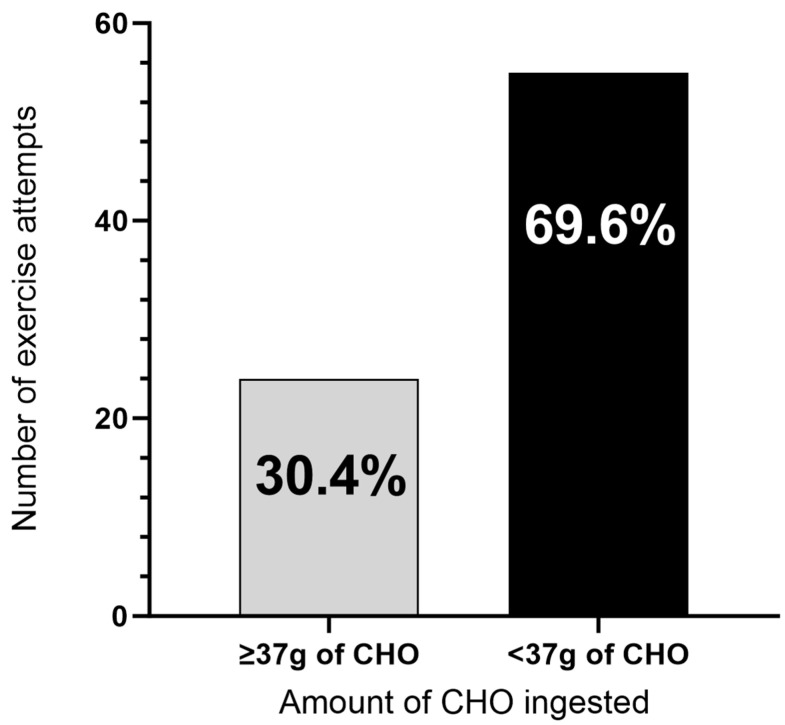
Comparison of the relative reporting of exercise attempts where participants consumed < vs. ≥37 g of carbohydrate pre-exercise. CHO = carbohydrate.

**Table 1 nutrients-16-01423-t001:** A summary of participant details who provided data on nationality (*n* = 108) and their gender, age, and weight (*n* = 104). Data on participant height were also collected for the subsequent calculation of BMI.

*n* = 104 (Males 27, Females 77)	Mean ± SD	Range
Age (y)	46 ± 15	18–82
Weight (kg)	86.7 ± 34.1	46–260
BMI (kg·m^2^)	30.4 ± 11.2	16.3–80.0
**Nationality**	**Number (*n* = 108)**	
Argentina	1	
Australia	15	
Belgium	1	
Canada	9	
Colombia	1	
Czech Republic	1	
Germany	3	
Hong Kong	2	
Ireland	2	
Italy	1	
México	1	
The Netherlands	3	
New Zealand	2	
Portugal	1	
Spain	4	
Sweden	1	
United States of America	44	
United Kingdom	16	

**Table 2 nutrients-16-01423-t002:** Collated responses from the number of days where continuous physical activity is performed (Questions 10) and the duration of that physical activity (Question 11) from participants self-reporting as IWMDs (*n* = 94) *.

	Question 11				
Question 10	15 min or Less	15–30 min	30–60 min	More than 60 min	Total
Once a week	5	5	2	1	13
2–3 times a week	4	14	12	4	34
More than 3 times a week	0	11	23	13	47
Total	9	30	37	18	94

* Of 104 survey responses in Q10, 10 participants selected that they do not undertake any continuous physical activity and therefore recorded no response for Q11.

## Data Availability

The original contributions presented in the study are included in the article/Appendix A, further inquiries can be directed to the corresponding author.

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
