# Peer review of "Carbohydrate Ingestion before Exercise for Individuals with McArdle Disease: Survey Evidence of Implementation and Perception in Real-World Settings"

_nutrients, 2024, doi:10.3390/nu16101423_

Round 1

Reviewer 1 Report

Comments and Suggestions for Authors

The manuscript presents and analyze the responses from 108 patients with McArdle disease to a survey focused on the self-perception of the benefits of ingesting carbohydrates prior exercise in real world settings. Although the focus and objective of this study is sound and interesting its elaboration, from my point of view, is clearly insuficient. The manuscript tries to take conclusions from mere subjectives answers, which per se is not pejorative, but lacks stronger design and analysis; first, the main conclusions are based on the following question to the patients: "Do you find the current.....releives/minimises your McArdle symptoms?" If the authors wanted to base a scientific work  and take conclusions based on perceptions from the patients ("Do you find...? is the question) they should help to constrain the degree of subjectivity in patients answers directing them to more objective facts: for example, "did your episodes of myoglobinuria were reduced with CHO ingestion"? How did you compare the effect of CHO inatke versus not CHO intake in RWS? Can you describe the type of exercise you performed? Normally how many minutes before exercise do you take carbohydrates?". Authors ahould also try to precise the concept "McArdle symptoms", and what exactly do the CHO relieve (or not): percieved pain? cramps? exertion? dilution of the second wind? myoglobinuria episodes?  Authors should also try to stratify the provided asnwers regarding the quantity/type of carbohydrate ingested, time prior exercise ingestion, time of the day in which exercise was performed? Stratification of results according to Martinuzzi phenotype scale of the different patients.  Why did the authors not used the accepted and validated questionnaires such as IPAQ, FSS, ADS-R, Borg of Rate of Perceived Pain scale, RPE-Rate of percieved exhertion? 

Author Response

We thank Reviewer One for their time and effort in providing comments for our manuscript. Please see the attachment for Responses to reviewer comments.

Reviewer 2 Report

Comments and Suggestions for Authors

Thanks for the chance to review this paper. This paper is about individuals with McArdle diseases. The aim of study was to determine if positive lab-based results for the ingestion of carbohydrate before exercise in laboratory studies are being effectively translated into practice and produce perceptions of the same positive outcomes in  real world settings.

My main query is that there should be incorporation of all 27 questions mentioned in the questionnaire to support and justify carbohydrate consumption variation in real world setting. Simply mentioning whether follow is being done or not is not interesting.For instance as a reader I will have several questions in mind as a layman. Few are listed below:

·       there may be variations in its prevalence or clinical presentation across different regions or populations. Investigating regional effects on MacArdle disease can provide insights into potential genetic, environmental, or demographic factors influencing the disease may be the reason of difference in followup of carbohydrate consumption. Classifying based on region may give some pattern and may worth explaining.

·       Studies may explore whether the consumption pattern is different  in males or females.

·       Timing about carbohydrate intake before exercise need to presented.

·       Which type of exercise the people are doing who are consuming >37g would be of interest to the readers.

·       The amount of carbohydrates needed before exercise may vary depending on factors such as the duration and intensity of the activity, individual metabolic rate, fitness level, and personal preferences. Work with a healthcare provider or nutritionist to determine the appropriate carbohydrate intake tailored to your specific needs. The questionnaire did not ask for any such question about their personal needs.

·       Stay adequately hydrated before exercise by consuming fluids along with carbohydrates. Proper hydration is crucial for maintaining optimal muscle function, preventing dehydration, and supporting overall exercise performance. Questionnaire didn’t include information about that.

Author Response

(The authors gave the same response as above.)

Round 2

Reviewer 1 Report

Comments and Suggestions for Authors

From my side, the review has been properly adressed

Author Response

Thank you for your time and consideration in providing actionable feedback to improve the manuscript quality. 

Best regards,